# Generating High-Resolution 3D Faces and Bodies Using VQ-VAE-2 with PixelSNAIL Networks on 2D Representations

**DOI:** 10.3390/s23031168

**Published:** 2023-01-19

**Authors:** Alessio Gallucci, Dmitry Znamenskiy, Yuxuan Long, Nicola Pezzotti, Milan Petkovic

**Affiliations:** 1Philips Research, 5656 AE Eindhoven, The Netherlands; 2Department of Mathematics and Computer Science, Eindhoven University of Technology, 5612 AE Eindhoven, The Netherlands

**Keywords:** 3D face synthesis, 3D body synthesis, artificial neural networks, generative modeling, 2D regular representation, autoencoders, autoregressive models

## Abstract

Modeling and representing 3D shapes of the human body and face is a prominent field due to its applications in the healthcare, clothes, and movie industry. In our work, we tackled the problem of 3D face and body synthesis by reducing 3D meshes to 2D image representations. We show that the face can naturally be modeled on a 2D grid. At the same time, for more challenging 3D body geometries, we proposed a novel non-bijective 3D–2D conversion method representing the 3D body mesh as a plurality of rendered projections on the 2D grid. Then, we trained a state-of-the-art vector-quantized variational autoencoder (VQ-VAE-2) to learn a latent representation of 2D images and fit a PixelSNAIL autoregressive model to sample novel synthetic meshes. We evaluated our method versus a classical one based on principal component analysis (PCA) by sampling from the empirical cumulative distribution of the PCA scores. We used the empirical distributions of two commonly used metrics, specificity and diversity, to quantitatively demonstrate that the synthetic faces generated with our method are statistically closer to real faces when compared with the PCA ones. Our experiment on the 3D body geometry requires further research to match the test set statistics but shows promising results.

## 1. Introduction

In the last two decades, the use and applications of virtual 3D models in the real world have risen exponentially. There are many reasons behind this, from the growth in computational power to the economic benefit of using a parametric model to simulate physical phenomena. Today, 3D models serve many fields, including animation of characters [1] and faces [2,3,4], recognition of expressions [5], face recognition [6], and inferring body shapes and measurement to be used, for example, in the clothing industry, for virtual try-on [7], or in the medical field to estimate fat distribution. However, the applications are often limited due to privacy and sensitive information constraints that reduce or block data sharing and aggregation from multiple sources. A common scenario arises when in a federated learning situation, multiple hospitals need to preserve the patients’ privacy while training machine learning models on the aggregate data. This problem is also accentuated by modern 3D scanners that operate with sub-millimeter accuracy, making a person identifiable from their 3D scan. At the same time, removing identity information, for example, using obscuring and decimation methods, can conflict with the modeling objectives and accuracy requirements.

To overcome such limitations, we propose to replace the original dataset with a synthetic replica and present a compelling solution to the generation of synthetic 3D models using a machine learning approach. As a first step, we registered a common reference 3D template into every scan, bringing all raw scans in full correspondence, as described in the seminal work of Blanz and Vetter [8] for faces and Allen et al. [9] for full bodies. Figure 1 shows a template for the face on the left and the one used for the full-body experiments on the right. Parameterization is insufficient to generate synthetic scans since the registered template has thousands of highly correlated vertices. Generation methods should consider this correlation by finding a low-dimensional de-correlated surface representation. Thus, Blanz and Vetter [8] used principal component analysis (PCA) as a data-driven approach to reduce the stacked array of thousands of vertex coordinates to a small number of de-correlated PCA scores. Then, to generate synthetic meshes, it is sufficient to sample the PCA scores as independent random variables. However, there are also disadvantages. In fact, not all the combinations of PCA scores result in a natural human shape, as illustrated in Section 6.

To overcome the drawbacks of PCA and similar linear methods, deep generative models based on convolutional neural networks (CNNs) were employed to capture more complex nonlinear interactions in the data. The current state-of-the-art advances in the field of geometric deep learning [10] leverage the power of CNNs by adapting them to work on meshes [11,12]. Graph convolutions, however, restrict the resolution and, therefore, the accuracy and extent of surface details of the 3D template. In our work, we decided to use an alternative approach and directly exploited CNNs by representing a registered 3D template on a regular 2D grid. Hence, a part of this work was devoted to the conversion from 3D shapes to 2D grids. We first considered a simpler case of the human facial surface using a 3D template with a CNN-friendly 2D grid structure, as shown in the left image of Figure 1, which can be represented as a 2D image, as shown in the left image of Figure 2. Second, we considered a general case of a complete human body, which cannot be naturally unwrapped into a 2D grid, as shown in the right image of Figure 1. One of the main contributions of this study is the proposal of a novel non-bijective method that represents a 3D shape as a sequence of projections on the 2D grid, as illustrated in the right of Figure 2. The advantage of this method is the simple creation of a 2D body template by means of rendering, which works with any parametrization of the 3D grid. The inverse 2D-to-3D conversion was achieved by the aggregation and regularization of information from multiple body projections. Once we obtained 2D representations of the scans, we applied CNNs to generate new 3D shapes. For our experiments, we used a relatively novel 2D image synthesis method employing a latent image representation given by a quantized variational autoencoder (VQ-VAE-2) [13], which was sampled using the autoregressive network PixelCNN [14] with the attention called PixelSNAIL [15]. Figure 3 depicts the full-body encoding with the two-level autoencoder on the top and the autoregressive full-body training and sampling at the bottom.

To evaluate our approach, we compared the synthetic 3D shapes generated with PCA and those obtained with the new CNN-based method. Subjectively, we found that our new method gave more natural 3D shapes, but due to the high variability of plausible 3D human shapes, the subjective evaluation was quite speculative. The challenge for an objective evaluation is finding metrics that can measure how the generated 3D shapes represent ‘human’ faces/bodies. Since we could not propose a metric for evaluating single scans, we evaluated how close the empirical distribution of the quantitative metrics computed over 250 synthesized faces/bodies compared with 250 real 3D shapes. The remaining paper content is structured as follows: In Section 2, we give an overview of the prior research, and, in Section 3, we describe the VQ-VAE-2 and PixelSNAIL models. In Section 4 and Section 5, we introduce the quantitative metrics used to evaluate our method and the conversion to 2D representations. The section on experiments describes the input data and different evaluations. Finally, we present the results and elaborate on them in the discussion and conclusion sections.

## 2. Related Works

Many works of research still rely on linear models [16] or multilinear models [17] due to their simplicity, or on the expansion of 3D morphable models [18]. Tran et al. [19] proposed a robust CNN-based approach to regress the PCA scores from pictures for face recognition and discrimination. In another work, multilinear models were used to transfer facial expressions and have the ability to animate faces [17]. While simple and easy to train, they do not take the input geometry into consideration. A review of the current methods’ regressing and sampling PCA scores is beyond the scope of this paper.

The 3D-to-2D representation methods. Many 2D representation methods originate from the solution of a rendering problem that relies on the so-called UV maps for the mapping of 2D textural images on 3D objects. UV maps, by definition, provide bijective maps from 3D mesh triangles to their images on the textural image. The UV maps for a face model can be created by warping the 3D templates with a regular grid of the facial surface; see, for example, Booth et al. [20] for a list of possible optimal implementations. In the current paper, we utilized a facial parametric model in which a 3D template already had a regular 2D grid.

Full-body UV maps can be coarsely divided into two classes: a single connected piece and a patchwork of multiple pieces. Regarding the single-piece solution, in 2002, Gu et al. [21] showed how to cut a surface and sample it over a regular 2D square grid, generating the so-called geometric image. While the work did minimize the artifacts along the cuts, it naturally cannot solve huge distortions when mapping the complex 3D shape of the body with arms and legs into a square grid. The majority of the recent prior research follows UV maps from the SMPL model [22], which has a patchwork of different body parts: a head, two palms, two arms, a torso, two feet, and two legs, each of which is economically mapped in its own place on the 2D grid so that the total area of background pixels is minimized. While the patchwork has more control over the surface area, it has many cuts through the surface, and every cut creates a challenge to obtain continuity in the generated vertex positions on both sides of the cut. Thus, in a more recent work, Zeng et al. [23] reverted to a single-piece solution where the geometrical distortions were reduced for the cost of an increased percentage of background pixels present in the UV map.

It is worth noting that the use of UV maps used in prior research for 3D–2D conversion is subject to licenses that constrain their commercial applications. While working for a company, we needed to find an efficient alternative that could also be implemented with minimal effort. Thus, in this work, we propose a novel non-bijective method that maps the 3D body into multiple 2D projections rendered from a set of camera views. This method minimizes the number of pieces and is generalizable to arbitrary 3D body templates.

Three-dimensional face generation with GANs. Abrevaya et al. [24] investigated the use of Wasserstein GAN [25] to generate novel 3D faces with the ability to control and modify their expression. However, in our work, we decided to directly map the input surface into a geometric image since we believe the MLP cannot efficiently handle the complexity of the shapes. In a study by Slossber et al. [26] and the extended work of Shamai et al. [27], similar to our work, a 3D shape was converted into a 2D regular representation by means of non-rigid registration techniques. In [28], Moschoglou et al. mapped the template T by using a cylindrical unwrapping method as introduced by Booth and Zafeiriou [20]. In our work, while using similar concepts, we did not use adversarial training.

Three-dimensional face generation with autoencoders. Apart from generative GANs models, recently, many works have overcome the limitations in linear modeling by using VAEs [29]. For example, Bagautdinov et al. [30] modeled the face using a multiscale approach for different frequencies of details. Fernandez et al. [31] exploited the power of a CNN-based encoder by coupling it with a multilinear decoder. In a study by Li et al. [32], a multicolumn graph convolutional network was designed to synthetize 3D surfaces. The authors first applied a spectral decomposition of the meshes and then trained multiple columns of graph convolutional networks. While these methods are similar to our approach, they also differ, as none of them uses a quantized autoencoder with an autoregressive network. Moreover, they do not convert data into 2D geometric images but often feed registered 3D scans directly.

## 3. VQ-VAE-2 with PixelSNAIL

The VQ-VAE model is introduced in [33], and it builds on the variational autoencoder (VAE) model [29,34] by generalizing ideas from classical image compression methods such as jpeg. Given a dataset of observations $\left\{x^{\left(1\right)},x^{\left(2\right)},\;\ldots ,x^{\left(N\right)}\right\}$, the goal of a VAE is to learn, without supervision, a lower-dimensional representation in terms of latent variables z. It is composed of an encoder E, which maps the input image into latent variables, and a decoder, which reconstructs the image from the compressed representation. In other words, the decoder network models the joint distribution $p(x|z)p\left(z\right)$, while the encoder models the posterior distribution q(z|x).

In the VQ-VAE framework, the prior distribution is based on K prototype latent vectors $\left\{e^{\left(1\right)},\;e^{\left(2\right)},\;\ldots ,e^{\left(K\right)}\right\}$ of dimension D, which quantizes the latent maps $E\left(x\right)$, generated by the encoder. There are exactly K different latent vectors to choose from, so each pixel on the latent maps is represented with the nearest quantizing vector. In Razavi et al. [13], the VQ-VAE-2 is presented, which is the upgrade of the VQ-VAE to include multiple hierarchical layers that provide different quantized codebooks at different hierarchies. The decoder then reconstructs the image using the latent maps conditioning the higher layers, which have smaller resolutions than the lower ones. In the original setup, the 24-bit image used as input had a 256 × 256 resolution, which was reduced to a 64 × 64 bottom map and a 32 × 32 top map with $K=512=2^{9}$ different quantizing vectors of D=64 dimensions. In [13], the authors presented a two-layer network trained on ImageNet [35] and a three-layer network trained on FFHQ [36] for generating high-resolution photorealistic facial images. In order to solve large-scale dependencies, which are usually difficult to capture using autoregressive decoders, Fauw et al. [37] successfully explored the possibility to use multilayer encoders, while in another research work, Williams et al. [38] used hierarchical quantized autoencoders for image compression purposes.

For new data generation, we applied the autoregressive model PixelCNN [14,39] with self-attention [40] called PixelSNAIL [15]. In this setup, the autoregressive model can efficiently model the prior distribution of latent codes, creating photorealistic synthetic images. The idea behind the PixelCNN model is to learn the conditional distribution of a given sequence of random variables. When applied to the latent space, the latent codes of the whole image are sorted from the top left to the bottom right to predict the next code value, which is a discrete probability distribution over K codes in an autoregressive fashion. In our example, the autoregressive model learns the joint distributions of the latent codes on the top layer and then the distribution of the bottom codes conditioned on the top codes. There are different options to generate new samples once the two models are trained. The main approach is to perform a sampling of the top space $c_{T}$ trained on a specific image class label and then sample the bottom space trained on the same label while conditioning it on the sampled top codes. In previous research, this approach was successfully applied for synthesizing photorealistic images of faces [13] and skin lesions [41]. In this paper, we applied the same method for the synthesis of 2D representations of 3D shapes.

## 4. Metrics for Quantitative Evaluation

Our goal was to provide a method that always generates realistic synthetic samples; to achieve this goal, we not only visually inspected the generated scans but also selected suitable metrics. The main idea was to prove that synthetic scans are statistically indistinguishable from a test set of original scans (which are excluded from the training). Before computing the metrics, the scans need to have identical parametrization corresponding to the 3D template. The identical parametrization enables a simple distance metric between a pair of scans, defined as an RMSE distance between the corresponding pairs of vertices, after the rigid alignment of one scan to another [42]. We employed two derivative metrics from the above RMSE distance, which are also used in the literature to evaluate synthetic scans.

The first metric is called diversity and was introduced by Abrevaya et al. [24], with the aim to produce a single number measuring the heterogeneity of a set of scans. Diversity is defined as the distance between a random pair of synthetic scans. In our work, we compared the empirical distribution of the diversity of 250 generated scans with the empirical distribution of the diversity of 250 original scans from the test dataset. The diversity D of a pair of scans with vertexes $v_{i}^{1}\in V^{1},\;v_{i}^{2}\in V^{2}$ and vertex weights $w_{i}\in W$ is defined as
$$D\left(V^{1},V^{2}\right)={\left[\frac{\sum_{i=1}^{N}w_{i}||v_{i}^{1}-v_{i}^{2}||{}_{2}^{2}}{\sum_{i=1}^{N}w_{i}}\right]}^{\frac{1}{2}}$$

The second selected metric is called *specificity* and is defined in [43] as a scan with minimal distance to the scans in the training dataset. Similar to the diversity distribution, we evaluated the empirical distribution of the specificity over 250 synthetic scans and compared it with the empirical distribution of the specificity in 250 original scans from the test dataset. Given *V* as the vertices of a synthetic scan, its specificity S is defined as
$$S\left(V\right)=\underset{t\in T}{\min}{\left[\frac{\sum_{i=1}^{N}w_{i}||v_{i}-v_{i}^{t}||{}_{2}^{2}}{\sum_{i=1}^{N}w_{i}}\right]}^{\frac{1}{2}}\;$$
where t is the index of the training set, N is the total number of vertices, and $v_{i}\in V,\;w_{i}\in W$ are the vertice weights.

## 5. From 3D to 2D Representations

Within this work, we considered two datasets of registered images: one corresponding to a face template with a regular grid of 128×128 vertices, shown on the left of Figure 1, and a full-body template with about 50 K vertices randomly placed over the body surface, shown on the right of Figure 1. The definition and registration of the templates are out of the scope of the current paper. Conceptually, we followed the method explained in [8,9] and morphed all scans by means of non-rigid registration methods [44,45]. A more detailed description of our parametric models is reported in [46].

Since the face template already had a grid structure, we only applied vertex-based normalization to map the range of input values into the interval $\left[0,\;1\right]$ and, therefore, to facilitate the follow-up processing with neural networks. The mapping to $\left[0,\;1\right]$ also facilitates the visualization of the normalized *xyz* facial data as *rgb* images, as illustrated in Figure 2. The range parameters for each grid vertex were retained for the denormalization of the synthetic images into 3D shapes.

In contrast to the face, the body template’s complex shape cannot be naturally unwrapped into a 2D grid. We, therefore, propose a novel non-bijective method that represents the 3D shape as a sequence of projections on the 2D grid. The advantage of this method is the simple creation of a 2D body template by means of rendering, which works with any parametrization of the 3D grid. In our experiments, we rendered three body views: from the front, from the back, and from the bottom; in the first two views, we morphed the arms down to save space on the rendered geometric image. The use of multiple views ensured that almost all the template vertices would be visible in at least one view, as shown in Figure 4 (left), where their displacement can be computed via the bilinear interpolation of 2D grid values.

The inverse conversion from 2D to 3D was achieved by aggregating bilinear interpolated vertex displacements from the views, followed by the regularization of the 3D shape using mesh Laplacian, as described in [47]. Thus, we found the vertex positions V by minimizing the quadratic cost function:$$V=\mathrm{argmin}\left(||G\cdot V-{P||}_{2}+\alpha ||L\cdot V-L\cdot {V_{0}||}_{2}\right)$$
where G is the sparse registration matrix for 2D grid points on the body, P are the de-normalized xyz vertex positions at the 2D grid points, α = 0.001 is the regularization parameter (for units defined in mm), L is the sparse matrix corresponding to discrete Laplacian, and $V_{0}$ indicates the average body vertices. Due to the use of the quadratic norm, it is easy to derive a closed-form solution for *V*, which gives us a mean vertex error of 0.14 mm, distributed as illustrated in Figure 4 (right).

Once the 3D bodies were mapped to the 2D grid, we applied the same normalization used in the case of faces to map the range of input values into the interval [0, 1]. We retained the range parameters for each grid point for use during the denormalization and conversion of the synthetic images into 3D shapes.

## 6. Experiments

In our experiments, we used 3D scans from two data sources: the SizeChina dataset of 3D head scans [48] and a CAESAR of full-body scans [49,50]. The above data gave us more than 5000 registered 3D face templates and more than 4000 registered 3D body templates (which were augmented to 50,000 registered bodies using ‘age’ and ‘weight’ growth models). We augmented the face dataset by performing a symmetric reflection over the y-axis, leading to more than 10,000 scans. We divided the dataset stratified according to participant id into 90% for training, 5% for validation, and 5% for testing, in both face and full-body experiments. We evaluated and computed the metrics only on the test set without considering the augmentations. For the sake of experimental reproducibility, we did not perform any other augmentation in training or test time.

Generation of synthetic data with PCA-based method. After registering the common templates, we encoded the information in 200 principal components (both for the head and full body). The description of our parametric model is presented in [46]. While the PCA analysis assumes that marginal coefficient distributions are close to Gaussian, it is more reasonable to follow a data-driven approach and sample from the empirical cumulative distributions of each coefficient. This approach brings multiple benefits: It is easy to implement and fast, and the sampled shapes will have marginal coefficient distributions statistically indistinguishable from the original data. Hence, we computed eCDF for all PCA-encoded scans separately. To sample a new scan, we simply sampled each score independently and then decoded the 3D scan from the scores.

Generation of synthetic data with CNN-based method. In our experiments, we focused on a two-layer VQ-VAE hierarchy with an input grid resolution of 128×128 (and relative latent maps with dimensions of 32×32 and 16×16) for the face and 256×256 (and relative latent maps with dimensions of 64×64 and 32×32) for the full body. We followed the approach described in [41] to find the best combination of K = [64, 128, 256 512] and D = [2, 4, 8, 16, 32, 64] and found that, according to reconstruction error, K = 512 was always better than smaller values. Conversely, for large enough K values, we noticed that a smaller dimension of D provides the best outcomes. Hence, we used D = 2 for our final VQ-VAE-2 model. We also reduced the batch size to 32, compared with the original implementation, for both the autoencoder and the autoregressive model. Concerning the autoregressive model, we used the original configuration for ImageNet apart from the batch size, 32 in our example, and the total number of epochs was 420 for both the top and bottom hierarchies. The autoregressive models’ validation accuracies in predicting the latent codes after 420 epochs were 0.87 for the top space and 0.91 for the bottom one when considering the face experiment. The full-body PixelSNAIL reached 0.33 for top accuracy and 0.72 for bottom accuracy. All the models were trained on PyTorch [51] with the same hyperparameters as in the original implementation (excluding the one explicitly mentioned above).

We present some randomly selected facial scans in Figure 5 (most of the figures displaying scans were rendered with MeshLab [52]): The top right shows the registered scans with the relative PCA-encoded versions on the top left; the bottom scans are synthetic images generated using our approach on the left and with PCA on the right. The PCA ones present more variability, or, in other words, more shape differences than the other sets. The synthetic scans generated with our approach, from a visual inspection, present similar shape variability to the original scans compared with the PCA synthetic scans. However, we believe that it is possible to spot the difference in shape distributions between the different sets. Figure 6 shows examples of synthetic scans generated with our method on the left and PCA synthetic scans on the right. From a visual inspection, the PCA scans seem less realistic, and, for example, in the torso area, artifacts seem to be present. Comparing scans through visual inspection is often not a trivial task and not an objective metric. Additionally, this is not enough since our approach may simply replicate or clone the original training data. We test these hypotheses in the following quantitative analysis proving that the synthetic scans are novel and different from the original training ones. In Figure 7, we show one example of how PixelSNAIL failed to produce correct geometric images for full bodies. This occurred for 0.5% of the generated scans, as shown by the intersection over the union plot in the right part of Figure 7. However, failure in generation can be easily detected by checking whether the background pixels were synthesized at the same places as in the original 2D template.

## 7. Quantitative Evaluation

We analyzed 2D representations for the registered raw vs. PCA-encoded vertices and computed empirical distributions for *specificity* and *diversity* metrics. The empirical distributions of metrics are shown in Figure 8 (face) and Figure 9 (full body). The face metrics show that our approach resulted in synthetic faces which were statistically close to the original scans in the test set, unlike the PCA-based method, which showed a flattened diversity distribution and higher specificity than the test set. The higher specificity confirmed the results of subjective evaluation, as shown in example scans in Figure 5, that the PCA-based method resulted in more extreme face shapes.

The full-body metrics did not confirm our subjective evaluation, as the distribution of *diversity* and *specificity* metrics for the PCA-based method seemed to be closer to the distribution in the test dataset selected from original scans. We can suggest the following reasons explaining this counterintuitive result: First, a full-scale body has more challenging 3D–2D mapping than a face with value discontinuity at the border of the projections. Quantitatively, this is already seen in the autoencoder and autoregressive having worse performances than those of the face experiment, which we believe is the main reason for the lower specificity and diversity. In fact, we believe that a less accurate autoregressive model led to the closer-to-average distributions of latent codes and 3D synthetic shapes. Another major point of discussion that will be analyzed in future work is to benchmark the effectiveness and quality of our augmentations prior to model training.

## 8. Discussion

While the experiments successfully generated realistic, high-resolution 3D faces and full bodies, we consider this only a first step in proving the validity of this approach. One main challenge is the lack of a clear, quantitative metric to judge whether a scan belongs to the “real” class (of faces or full bodies). These proposed diversity and specificity metrics may not be enough to capture all the relevant shape information. Moreover, the full-body experiment still requires further investigation in many directions. For example, we did not fine-tune any network hyperparameters, and we did not optimize UV mapping and augmentation for the current task at hand. In fact, the generation of full-body 2D representations was required for different applications. We suggest that future research consider optimizing the number of projections and their position while minimizing the background pixels to facilitate NN training. Moreover, we believe further realistic augmentations would impact and consolidate the results of both experiments. Nevertheless, we also noticed that the current set of facial scans is enough to achieve the desired outcome of statistical indistinguishability from the test set.

A natural extension of our approach could be to combine the 3D shape synthesis with the photorealistic textural synthesis from [13], which can be added as rgb channels to xyz channels within the 2D representations.

## 9. Conclusions

In this paper, we presented a novel approach that can generate high-resolution synthetic 3D scans for the face and the full body. While tackling the more complex full-body geometry, we presented a new non-bijective way of creating a 2D representation of a 3D template by using multiview projections. The advantage of this method is that it is agnostic to the shape of the 3D template and can be adapted to any 3D template: foot, hand, torso, etc. The remaining challenge (and the subject of optimization) is finding the best set of camera views while minimizing the overlap and the number of invisible vertices.

Another relevant strength of our approach is that it does not require the parametrization of a 3D face/body model and can be directly applied to registered templates. This leads to the possibility to generate scans outside of the sub-space of linear PCA. However, the major contribution of our work is that our method strictly outperforms the classical approach of linear PCA. In fact, the generated high-resolution facial scans are superior in terms of the specificity and diversity metrics by a significant margin, as shown in the quantitate evaluation. Moreover, our method also outperforms the PCA-based one when training it over the PCA-encoded scans in the face experiment. Further work is required to ensure that the same conclusions also hold for the full-body surface.

To conclude, we demonstrated that is possible to apply state-of-the-art CNNs for the generation of realistic high-resolution 3D scans by reducing the problem to 2D representations. We showed that the combination of VQ-VAE-2 with PixelSNAIL, which was previously used for the generation of realistic facial images and skin lesions, is also applicable to 3D meshes when representing them as images. Finally, we showed that our approach outperforms PCA-based sampling via quantitative and qualitative analyses of synthetic scans.

## Figures and Tables

**Figure 1 sensors-23-01168-f001:**
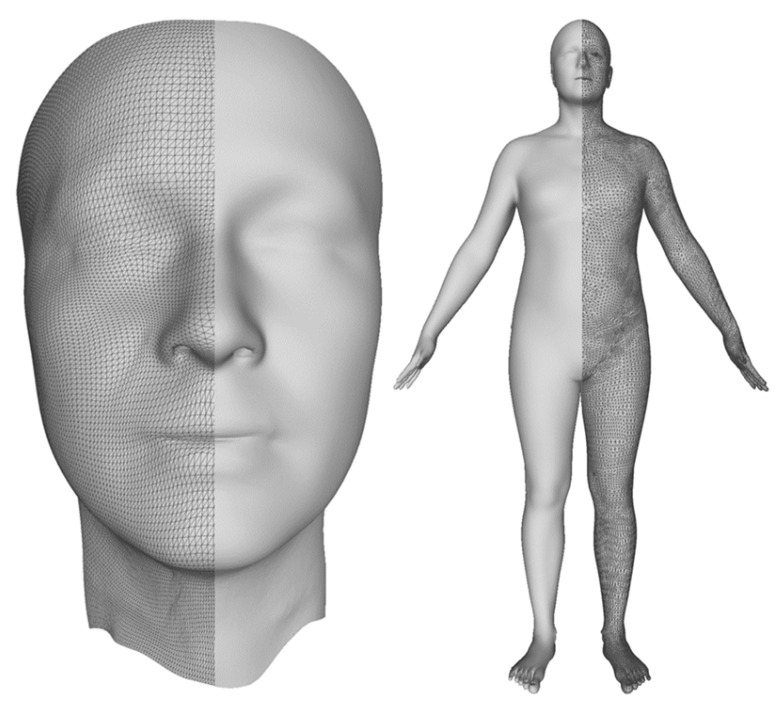
Templates. The average face template is on the left, and the average full-body template is on the right. The face has a regular (triangular) mesh grid as opposed to the less structured, but still simple, full-body mesh. The figure, for visualization purposes, shows templates with half-rendered surfaces and half as meshes.

**Figure 2 sensors-23-01168-f002:**
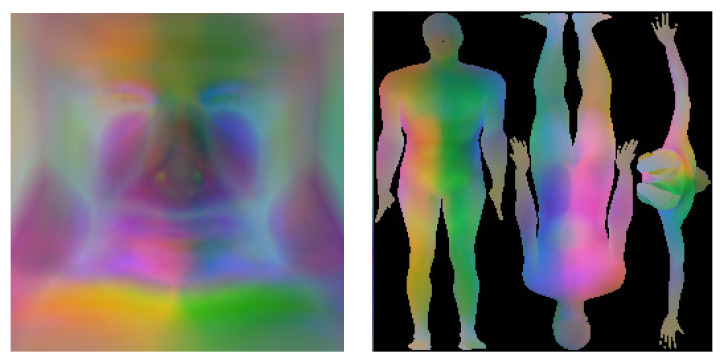
Geometric images. Examples of 128 × 128 geometric images for the facial template (**left**) and 256 × 256 for the body template (**right**). The left body image shows a frontal view, the middle shows a rear view, and the right is viewed from the bottom.

**Figure 3 sensors-23-01168-f003:**
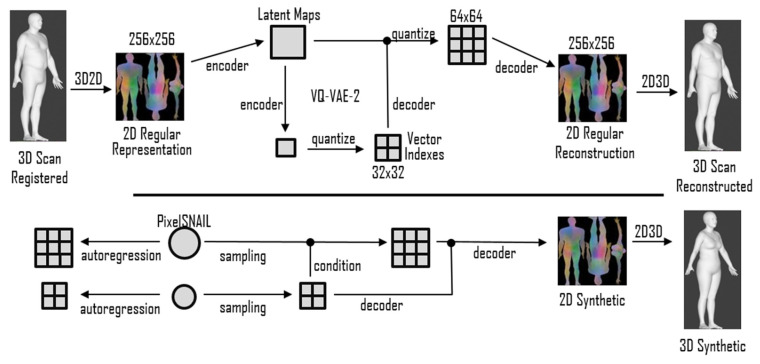
Method’s flowchart for a 3D body example. The registered 3D scans were first converted into a regular 2D image to feed the VQ-VAE-2 autoencoder. The PixelSNAIL later learned a prior over the latent space, sampled novel synthetics codes, and decoded them into new geometric images and, subsequently, to 3D scans.

**Figure 4 sensors-23-01168-f004:**
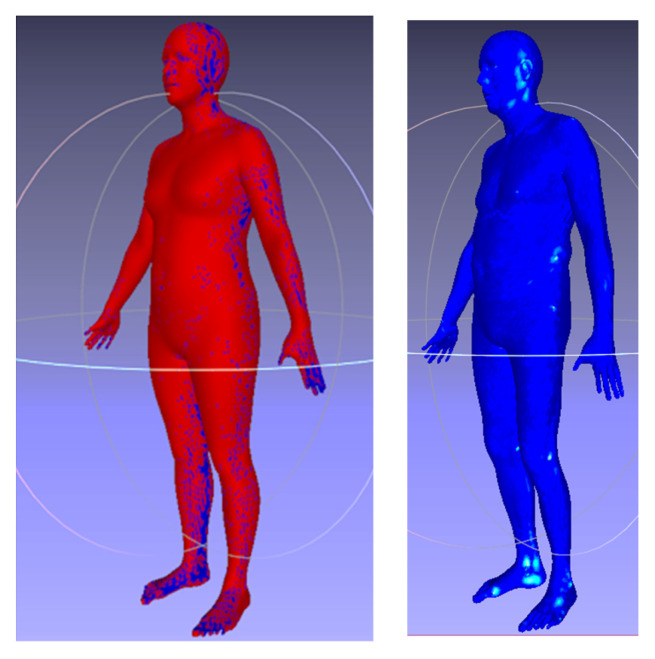
The 2D–3D reconstruction accuracy. The left image shows visible (red) vs. invisible (blue) vertexes in the template body. The right image shows the distribution of the reconstruction error, according to the ‘jet’ colormap (blue = 0.0 mm and red = 1.00 mm) in a sample body.

**Figure 5 sensors-23-01168-f005:**
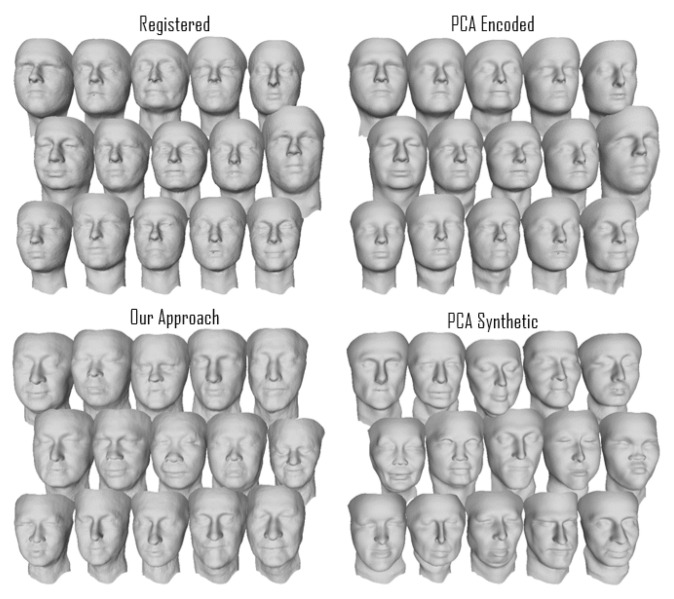
Example of facial scans (without selection). A batch of registered scans (**top left**), the same scans encoded (**top right**), synthetic scans generated with our approach (**bottom left**), and PCA synthetic scans (**bottom right**).

**Figure 6 sensors-23-01168-f006:**
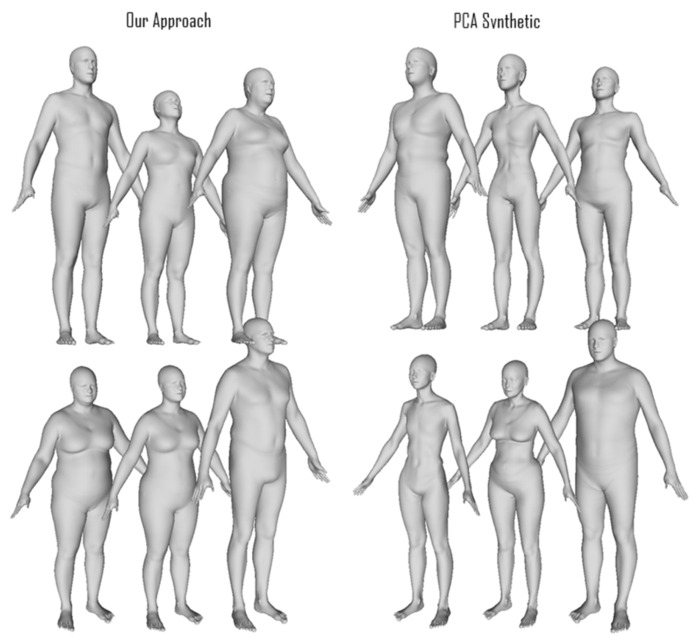
Example of full-body scans. The three columns on the left are sampled using our approach, and the three columns on the right are samples generated using the PCA-based approach. Subjectively, our approach provides realistic scans more often than the PCA-based approach, where we observe extreme shape features and often bodies with mixed gender that are not representative of the input dataset.

**Figure 7 sensors-23-01168-f007:**
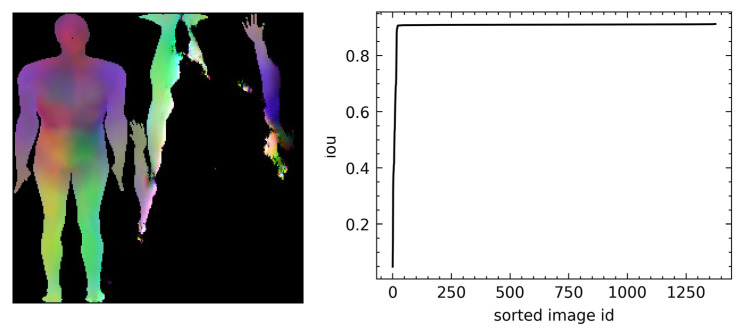
An example in which the PixelSNAIL failed to produce a correct geometrical image is on the left. The right is the distribution of intersection over unions over the segmented UV body mask.

**Figure 8 sensors-23-01168-f008:**
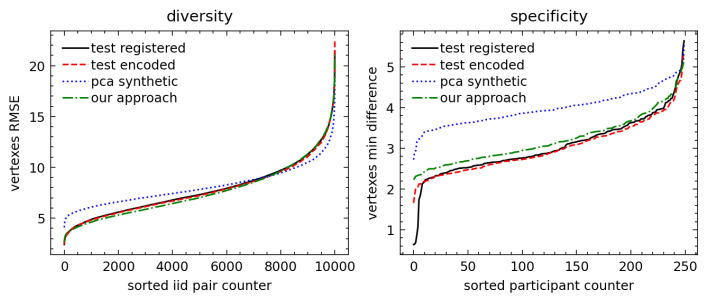
The face metrics. On the left, the diversity is plotted for each i.i.d pair and shows that the PCA distribution is “flatter”, as excepted when using the linear method. On the right, the specificity (the minimum distance versus the training set was kept) shows that our approach is much closer to the test set. Moreover, the specificity also proves that we did not replicate the input training scans since the minimum distance was markedly above 0 mm—with our approach above 2 mm.

**Figure 9 sensors-23-01168-f009:**
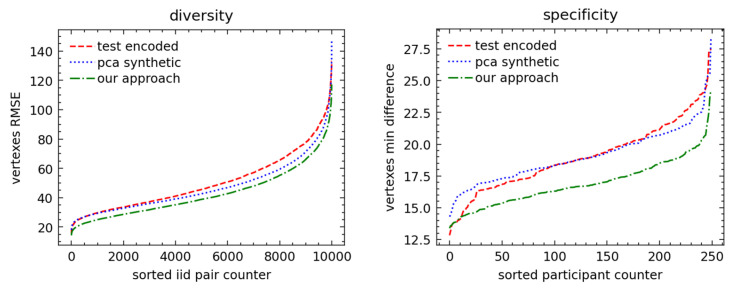
The full-body metrics. The full-body metrics do not show promising results as in the facial geometry, and further experiments are necessary to reach the desired quality of scans. Both diversity and specificity for our approach are lower than test set distributions, thus allowing us, for example, to calibrate the model and increase the variability of the scans by raising the sampling temperature and, therefore, increasing the entropy, diversity, and specificity of the scans.

## Data Availability

The CAESAR dataset is available at http://www.shapeanalysis.com/CAESAR.htm while the SizeChina dataset is available at https://web.archive.org/web/20070701104704/http://www.sizechina.com/html/intro.html.

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
