# Peer review of "Generating High-Resolution 3D Faces and Bodies Using VQ-VAE-2 with PixelSNAIL Networks on 2D Representations"

_sensors, 2023, doi:10.3390/s23031168_

Round 1

Reviewer 1 Report

The paper presents an approach to procedurally synthesise 3D full body and face shapes from a 2D image representation. The used CNN contains a variational autoencoder (VQ-VAE-2) and new models are generated using a variation of the autoregressive PixelCNN (with attention), named PixelSNAIL.

The paper is well written (clear and objective). The English language and writing style are fine. 

A very minor issue is that there are a few considerations/choices related to the research environment (Philips Research, where the authors are affiliated, according to the paper). It doesn’t sound very scientific and I would suggest rephrasing it (but this is a small issue).

The main issue is that the technique seems very incremental. The results are not bad, but it is important to clarify what the contributions of the paper are. The authors should highlight more what they added to the core of the method (CNN), and why they made the architectural choices presented in the paper.

Furthermore, the proposed approach was well-evaluated against PCA (calculate PCA over encoder parameters and sample the ”valid space”). The authors used well-known metrics, showing that this technique is better than PCA. However, the proposed method was compared with a very old work (from 1999). The result should be also compared with something new, for example, against other architectures used to generate other types of 3D models, or other more recent related approaches (not necessarily for body generation, but that could be applied to this context) the authors consider relevant.

Finally, some future works mentioned in the paper could be incorporated to increase the relevance of this work.

Reviewer 2 Report

The Article “Generating high-resolution 3D faces and bodies using VQ-2 VAE-2 with PixelSNAIL networks on 2D representations” is dedicated to tackle the problem of 3D face and body synthesis by reducing 3D meshes to 2D image representations.

Undoubtedly, the creation of parametric models of 3D living objects is an interesting task that has applications in various fields. The authors show the results of their proposed novel non-bijective 3D-2D conversion method and show interesting results.

However, the style of presentation does not allow to understand some points. For example, the authors in the abstract state "The experiment on 3D body geometry requires further research to match the test set statistics but shows promising results"(23). But neither in the Discussion nor in the Conclusions do the authors specifically decipher what “promising results” they were talking about.

There are a number of inaccuracies in the Article, such as "Figure 3 shows on the left the template for the face and on the left the one used for the full body experiments"(55-56). It is clear that the second “left” should be replaced with “right”, but this is unpleasant to read. Or the concept of "temperature 1.0." (359).

I'm not talking about the fact that the article first shows Figure No. 9, and after it, then Figure No. 8.

In addition, authors often allow the expression of their opinion, which is not supported by anything, for example: While in this application we assume that asymmetries are normally distributed on the left and on the right of face we do not know if this is true (302), ore " We still believe these augmentation does not hamper the results of the approach” (303).

All this does not affect the results obtained by the authors, but it interferes with their proper understanding and even raises the question to the result trus.

Therefore, I think that the authors should once again carefully rework the article and eliminate all the shortcomings. I would like to wish the authors to be more attentive to readers. To do this, both in the discussion and in the conclusions, firstly, to confirm what is stated in the annotation. Secondly, the statements made in the conclusions: "However, 408 the major contribution of our work is that our method strictly outperforms the linear PCA classical approach and generates realistic high-resolution scans" (409), need to be confirmed by quantitative results - by how much superior and on what scale?

I think that after processing this article may well be published in Sensors.

Reviewer 3 Report

1- English needs to be improved, we can find some grammatical mistakes and punctuation problems, and the author needs to go over the introduction and related work section again. 

2- Recent citations are missing so kindly add related papers from 2022.

3- The introduction part is too lengthy and some of the paragraphs of the introduction part can be accommodated into related works or methodology.

4-Last paragraph of related work where the author mentioned some related and relevant work so the weakness of cited research work can be mentioned. Elaborate on the significance of quantized autoencoder with the autoregressive network.

5- Highlight the novelty/contributions of the proposed work in the introduction part.

6- Discussion on the attained results is so weak and limited. Discuss and compare the achieved qualitative and quantitaive results with the state of the art.

7- In the experiment part, why only the reflection augmented scheme is used, and why not others?

8- Page#08, name the other realistic augmentation schemes that can further impact.

9-Which CNN is used to generate the synthetic data and define the K and D.

10- Page#09, the author claims that obtained qualitative results are better than PCA generated and closer to the original one but How? In terms of texture or shape? 

11- It is just a suggestion to add some more quantitative or exploration results about your findings during the research work. 

Round 2

Reviewer 3 Report

Although only 50% of my recommendations/suggestions are accommodated.

Still, the justifications of experiments and discussion are weak. It would be great if the author compare recently published research in the context of qualitative observations. Anyway, accepted.

Author Response

Thank you for your suggestion. We agree that comparing recently published work will improve the paper. Adding qualitative comments on the results of the experiments not backed up by quantitative experiments might reduce the quality of the paper. It is highly debatable how to judge if a scan is realistic and does not replicate the training scan. Moreover, the recent research significantly differs from our approach (e.g. by generating scans with pose variations in scenes, by generating lower resolution faces, including emotions).